# Genotype Frequency of *HLA-B*58:01* and Its Association with Paraclinical Characteristics and *PSORS1C1* rs9263726 in Gout Patients

**DOI:** 10.3390/diagnostics15162114

**Published:** 2025-08-21

**Authors:** Hien Thu Nguyen, Ha Thi Bui, Yen Thi Thu Hoang, My Ha Hoang, Manh Duc Ngo, Mai Hoang Nguyen, Thuy Thi Thanh Nguyen, Nhuan Tien Ngo, Quang Viet Nguyen

**Affiliations:** 1Department of Biochemistry, Faculty of Foundation Medicine, Thai Nguyen University of Medicine and Pharmacy, Thai Nguyen University, Thai Nguyen 250000, Vietnam; 2Department of Biotechnology, Faculty of Natural Science and Technology, Thai Nguyen University of Sciences, Thai Nguyen University, Thai Nguyen 250000, Vietnam; 3Deparment of Biology, Thai Nguyen Specialized School, Thai Nguyen 250000, Vietnam; 4Outpatient Department, Thai Nguyen National Hospital, Thai Nguyen 250000, Vietnam; 5Department of Occupational Health and Environmental Health, Faculty of Public Health, Thai Nguyen University of Medicine and Pharmacy, Thai Nguyen University, Thai Nguyen 250000, Vietnam

**Keywords:** *HLA-B* gene, *PSORS1C1* gene, *HLA-B*58:01*, rs9263726 SNP, gout patients, paraclinical characteristics

## Abstract

**Background/Objectives**: The *HLA-B*58:01* allele is strongly linked to severe cutaneous adverse reactions (SCARs) during allopurinol treatment, and it has been associated with the A allele of *PSORS1C1* rs9263726 (G>A). Paraclinical characteristics of gout are indicative of associated comorbid conditions. This study investigated the genotype frequency of *HLA-B*58:01* and its association with paraclinical characteristics and *PSORS1C1* rs9263726 in gout patients from Northeast Vietnam. **Methods**: A total of 133 unrelated gout patients were randomly recruited by the clinician. BioEdit sequence alignment editor version 7.2.5 software (Raleigh, Raleigh, NC, USA) was used for the analysis of nucleotide sequence data of *HLA-B* gene alleles from the IPD-IMGT/HLA Database, which showed that the *HLA-B*58:01* allele can be distinguished from reference and other alleles by specific nucleotide positions: 387C, 379C, 368A, 355A, and 353T (in exon 3); and 319C, 285G, and 209A (in exon 2). *HLA-B*58:01* and *PSORS1C1* rs9263726 genotypes were identified using Sanger sequencing of PCR products, analyzed with BioEdit software, and verified using the NCBI dbVar database. Statistical analyses were performed using SPSS version 25.0. **Results**: Our study revealed a significant age difference between male and female gout patients (*p* < 0.001). Male gout patients had an average age of 51.44 ± 14.59 years, whereas female gout patients were notably older, with an average age of 70.33 ± 10.64 years. Positive correlations were observed between platelet count, serum creatinine, and uric acid levels (r = 0.174, *p* = 0.045; r = 0.195, *p* = 0.025) in male gout patients, while only high-density lipoprotein cholesterol showed a statistically significant negative correlation with uric acid levels (r = −0.885, *p* = 0.002) in female patients. The *HLA-B*58:01* allele frequency among study subjects was 6.02%, with 12.03% being heterozygous individuals (**X/HLA-B*58:01*, N = 16). The *HLA-B*58:01* allele was not detected in female gout patients. White blood cell counts were significantly higher in male gout patients with the **X/HLA-B*58:01* genotype compared to those with the **X/*X* genotype (*p* = 0.018). The A allele frequency of *PSORS1C1* rs9263726 was 7.89%, and the heterozygous mutant genotype *PSORS1C1* GA had a frequency of 15.79% (N = 21). Among the **X/*58:01* carriers, 4.51% had the GG genotype, and 7.52% had the GA genotype at *PSORS1C1* rs9263726. **Conclusions**: Our study showed that the *HLA-B*58:01* allele was not detected in female gout patients. White blood cell counts differed significantly between the **X/HLA-B*58:01* and **X/*X* groups in male gout patients. The A allele of *PSORS1C1* rs9263726 was not consistently associated with *HLA-B*58:01* and was not a reliable marker for its detection in this study population.

## 1. Introduction

Gout is a disease caused by a disorder of purine metabolism, primarily characterized by hyperuricemia. Common complications of gout include bone fractures, kidney stones, and serious stroke conditions that can lead to disability or even death [1]. Hyperuricemia is the main cause of gout; therefore, patients are typically treated with prescription medications to manage this condition. Acute gouty arthritis is usually treated with non-steroidal anti-inflammatory drugs (NSAIDs), colchicine, or a combination of both, while long-term management focuses on medications that lower blood uric acid (Uri) levels [2,3,4,5]. Allopurinol, a xanthine oxidase inhibitor, is often the first-line treatment due to its affordability, convenient dosing schedule, and well-established long-term efficacy. However, severe cutaneous adverse reactions (SCARs) occur in 2–3% of patients taking allopurinol [6], with a mortality rate as high as 26% [7].

Several single-nucleotide polymorphisms (SNPs) in the coding regions of genes associated with the drug response in gout treatment have been identified, including *CYP2C9*, *HLA-B*, and *G6PD* [2,3,4,5]. Among these, the *HLA-B*58:01* allele (also referred to as *HLA-B*58:01:01* or simply *HLA-B*58:01*) of the *HLA-B* gene is strongly associated with SCARs during allopurinol treatment. As such, *HLA-B*58:01* is considered a predictive marker for severe skin hypersensitivity. This allele is codominant, meaning that an individual needs only one copy to be at increased risk [4]. Globally, *HLA-B*58:01* has been shown to confer susceptibility to allopurinol-induced SCARs in populations from Taiwan [7], Thailand [8], Japan [9], Korea [10], Malaysia [11], and Australia [12]. According to the EMBL-EBI database (European Molecular Biology Laboratory–European Bioinformatics Institute, EMBL-EBI), the *HLA-B* gene has over 200 known alleles, and *HLA-B*58:01* differs from the reference sequence by 58 nucleotide positions “https://www.ebi.ac.uk/ipd/imgt/hla/alleles/ (accessed on 15 August 2023)”.

To date, *HLA-B*58:01* has been detected using several methods, including a microsphere-based array genotyping platform with sequence-specific oligonucleotide probes [13], PCR-SSR (targeting exons 2 and 3), PCR-RFLP (based on the rs9263726 SNP in the *PSORS1C1* gene, G→A), and real-time PCR using TaqMan or SYBR Green probes [14] as well as real-time PCR with sequence-specific amplification (e.g., from Pharmigene, Taipei). Among these, rs9263726 has been reported as a potential surrogate marker for detecting *HLA-B*58:01*. However, the strength of this association varies across populations [15]. For instance, *PSORS1C1* rs9263726 is not tightly linked to *HLA-B*58:01* in the Australian [16] and Thailand populations [17], it shows a consistent linkage in the Japanese population [9], and it yields inconsistent results across different Chinese subgroups [18,19]. In 2015, *HLA-B*58:01* was predicted to be clinically associated with a high incidence of allopurinol-induced SCARs in Vietnamese patients [20]. Subsequently, Nguyen et al. [21] confirmed a strong association between *HLA-B*58:01* and SCARs in Vietnamese gout patients, making Vietnam the third most affected population worldwide after Taiwan [7] and Thailand [8]. Furthermore, their study suggested that *PSORS1C1* rs9263726 could serve as a surrogate marker for detecting *HLA-B*58:01* in the Vietnamese population, offering a cost-effective and simplified alternative to genetic screening.

On the other hand, it is important to understand that gout and hyperuricemia are not simply conditions that trigger painful joint attacks; rather, they are systemic metabolic disorders associated with a wide range of comorbidities, including cardiovascular disease, chronic kidney disease, diabetes, insulin resistance, fatty liver disease, osteoarthritis, as well as respiratory and ocular disorders [22,23]. The correlation between Uri and creatinine (Cre) likely reflects impaired renal clearance, a known contributor to elevated Uri levels [24,25]. Moreover, some *HLA* alleles are associated with the estimated glomerular filtration based on serum Cre levels [26]. Meanwhile, the association between Uri and platelet (PLT) count may indicate a link between gout and advanced atherosclerosis and could serve as a potential predictor of acute myocardial infarction [27,28]. In addition, higher total fat mass, trunk fat mass, and the trunk-to-leg fat mass ratio were significantly associated with increased levels of blood glucose (Glu), triglycerides (TG), and blood pressure while showing an inverse association with high-density lipoprotein cholesterol (HDL-C) levels [29].

According to the allele frequency database (The Allele Frequency Net Database, “http://www.allelefrequencies.net/hla6006a.asp (accessed on 15 August 2023)” and several studies on *HLA-B*58:01* in the Vietnamese population, it has been shown that Vietnamese individuals have a high prevalence of the *HLA-B*58:01* allele, ranging from 6.0% to 8.42% [13,30,31]. Therefore, in this study, we aimed to determine the genotype and allele frequencies of *HLA-B*58:01* in gout patients living in the northern region of Vietnam and to investigate its association with paraclinical characteristics and *PSORS1C1* rs9263726 using the Sanger sequencing method, with the goal of developing testing strategies to support treatment in Vietnam.

## 2. Materials and Methods

### 2.1. Subjects

The subjects were 133 unrelated gout patients enrolled randomly between January 2023 and June 2024 at Thai Nguyen National Hospital, Thai Nguyen, Vietnam (aged 26 to 88 years). Gout was diagnosed by clinicians based on etiology, medical history, clinical manifestations, complications, laboratory tests, imaging, and histological findings [32]. The aim of this study was explained to all participants, and informed consent was obtained from each subject, with strict protection of their privacy. This study was approved by the Human Ethics Committee of Thai Nguyen National Hospital (Thai Nguyen, Vietnam), Ministry of Health of Vietnam (Hanoi, Vietnam) (Approval No. 882/HDDD-BVTWTB).

### 2.2. Paraclinical Characteristics Analysis of Subjects

Analyses of paraclinical characteristics were performed using standard operating procedures (SOPs) at Thai Nguyen General Hospital, following instructions from the Ministry of Health of Vietnam, as described by Hoang et al. [33].

### 2.3. DNA Extraction, PCR Direct Sequencing, and Genotype Analysis

Total genomic DNA was extracted as described by Hoang et al. [33,34]. Primers for the PCR and sequencing of *HLA-B* (exons 2 and 3) and *PSORS1C1* (exon 3) were designed based on reference sequences in GenBank with accession numbers NG_023187 and NG_021348, respectively (Table 1). All primers were synthesized and supplied by PHUSA Biochem, Can Tho, Vietnam. PCR and Sanger sequencing methods of exons 2 and 3 of the *HLA-B* gene and exon 3 of the *PSORS1C1* gene carrying SNP rs9263726 were performed according to a previous report [33,34]. The thermal cycling conditions for amplifying the exon 3 fragment of the *PSORS1C1* gene and exons 2 and 3 of the *HLA-B* gene were as follows: an initial denaturation at 95 °C for 3 min, followed by 35 cycles of 95 °C for 45 s, 58–59 °C for 30–45 s, and 68 °C for 30–45 s, with a final extension at 72 °C for 5 min.

### 2.4. Method for Identifying the HLA-B*58 Allele and PSORS1C1 rs9263726 in Gout Patients

Analysis of nucleotide sequence data of *HLA-B* gene alleles from the EMBL-EBI Database “https://www.ebi.ac.uk/ipd/imgt/hla/alleles/ (accessed on 15 August 2023)”. using the BioEdit sequence alignment editor version 7.2.5 (Raleigh, NC, USA), showed that the *HLA-B*58:01* allele can be distinguished from the reference sequence (HLA00132.1) and other alleles based on the nucleotide sequences of exons 3 and 2 of the *HLA-B* gene. In our study, exon 3 was sequenced from 133 patient samples. Then, samples potentially carrying the *HLA-B*58:01* allele were selected based on the presence of nucleotides 387C, 379C, 368A, 355A, and 353T, and exon 2 was subsequently sequenced. A patient is considered to carry the *HLA-B*58:01* allele if exon 2 contains the nucleotides 319C, 285G, and 209A. rs9263726 was identified based on nucleotide sequencing of exon 3 of the *PSORS1C1* gene (110 G>A).

The genotypes of *HLA-B*58:01* and SNP rs9263726 were detected using BioEdit sequence alignment editor version 7.2.5 software and the database of human genomic structural variation (dbvar) of NCBI data.

### 2.5. Statistical Analysis

The frequencies of alleles and genotypes and paraclinical characteristics testing results were obtained using counting methods. The differences between the allele and genotype frequencies in this study and in other reports were considered statistically significant when *p* < 0.05. All statistical analyses were performed using SPSS version 25.0 software (Armonk, New York, NY, USA).

## 3. Results

### 3.1. Age, Gender, and Paraclinical Characteristics of Subjects

The age, gender, and paraclinical characteristics of 133 gout patients residing in Northeast Vietnam are presented in Table 2 and Table 3.

Table 2 shows that the majority of the gout patients were male, accounting for 93.2% (124/133), while females comprised only 6.8% (9/133). The average age of the study population was 52.71 ± 15.09 years. When analyzed by gender, the male gout patients had an average age of 51.44 ± 14.59 years, whereas the female patients were notably older, with an average age of 70.33 ± 10.64 years. Interestingly, all patients in the ≤40 group were male, and most female patients (77.8%) were in the ≥60 group. A statistically significant difference in age was observed between the male and female patients (*p* < 0.001), indicating that female gout patients tend to be older than their male counterparts.

Regarding the correlation between uric acid (Uri) concentration and various paraclinical characteristics, the patients were subdivided into total patients (N = 133), male patients (N = 124), and female patients (N = 9). For total patients, a weak but statistically significant positive correlation was observed between PLT count and Uri concentration (r = 0.174, *p* = 0.045) as well as between serum Cre levels and Uri concentration (r = 0.195, *p* = 0.025). Among male patients, significant correlations were also seen for PLT (r = 0.202, *p* = 0.024), Ure (r = 0.217, *p* = 0.016), and Cre (r = 0.215, *p* = 0.017). Notably, in female patients, only HDL-C showed a statistically significant negative correlation with Uri levels (r = −0.885, *p* = 0.002). These results suggest a potential gender-specific relationship between Uri and certain biochemical parameters, especially renal function markers and lipid metabolism (Table 3).

### 3.2. Genotype and Allele Frequencies of HLA-B*58:01 and PSORS1C1 rs9263726

Figure 1 presents representative sequencing chromatograms of study samples showing nucleotide positions in the *HLA-B* gene, marked to determine the *HLA-B*58:01* allele, and the SNP rs9263726 in the *PSORS1C1* gene, highlighting various single-nucleotide polymorphisms (SNPs) identified at specific nucleotide positions within the gene of interest. Panels A–B show multiple sequence alignments, with arrows indicating the precise locations of nucleotide substitutions in the *HLA-B* gene to determine *HLA-B*58:01*. Specifically, Panel A displays sequencing results of exon 3 of the *HLA-B* gene from five samples (1–5), highlighting nucleotide positions 387, 379, 368, 355, and 353. Samples 3 to 5 show the characteristic nucleotide pattern (387C, 379C, 368A, 355A, and 353T), indicative of *HLA-B*58:01*, while samples 1 and 2 do not. Panel B shows sequencing chromatograms of exon 2 for the same or corresponding samples, focusing on positions 319, 285, and 209, which are also used to confirm the presence of *HLA-B*58:01*. Panel C specifically presents the genotyping results for the rs9263726 SNP within the *PSORS1C1* gene. It shows chromatograms corresponding to the three different genotypes, namely GG, GA, and AA, with arrows pointing to the polymorphic site. These chromatograms demonstrate distinct peak patterns that differentiate the homozygous wild-type (GG), heterozygous (GA), and homozygous variant (AA) genotypes.

Data on genetic polymorphisms in two gene regions, namely *HLA-B* exons 2 and 3 and *PSORS1C1* exon 3, are presented in Table 4. For the *HLA-B* gene, three genotype groups were observed: **X/*58:01* (heterozygous genotype, 16 individuals, 12.03%)*; *58:01/*58:01* (homozygous mutant genotype, 0 individuals, 0%); and other genotypes of the *HLA-B* gene (**X/*X*), with 117 cases, accounting for 87.97%. The allele frequencies indicate that **X* was present in 93.98% of cases and that **58:01* was present in 6.02% of cases. For rs9263726 of the *PSORS1C1* gene, the homozygous wild-type genotype *PSORS1C1 GG* had the highest frequency at 84.21% (N = 112), and the heterozygous mutant genotype *PSORS1C1 GA* had a frequency of 15.79% (N = 21). The allele frequencies show that *G* is predominant at 92.11%, while *A* is present at 7.89%.

### 3.3. Association Between HLA-B*58:01 and Paraclinical Characteristics

The correlation between HLA genotypes (**X/*X* vs. **X/*58:01*) and various paraclinical characteristics in gout patients is presented in Table 5. Among all patients, WBC counts showed a statistically significant difference between genotypes, with higher values in **X/*58:01* carriers (12.351 ± 9.036 × 10^9^/L) compared to **X/*X* (9.649 ± 3.086 × 10^9^/L), *p* = 0.018. The same significant difference was observed in male patients (*p* = 0.018). All other parameters, including RBC, HGB, HCT, NE, LYM, PLT, Glu, Ure, Cre, Uri, TC, TG, HDL-C, and LDL-C, showed no statistically significant differences between genotypes in either the total group or male subgroup (*p* > 0.05). These results suggest that the presence of the *HLA-B*58:01* allele may be associated with elevated WBC levels, potentially indicating an altered inflammatory or immune response in gout patients carrying this allele.

### 3.4. Association Between HLA-B*58:01 and PSORS1C1 rs9263726

The distribution of combined genotypes for the *HLA-B* gene and the *PSORS1C1* gene polymorphism rs9263726 among the 133 individuals is shown in Table 6. The majority of individuals (79.7%) had the **X/*X* genotype for *HLA-B* and the *GG* genotype for *PSORS1C1*. A smaller proportion (8.27%) had the **X/*X* and *GA* genotype combination. Among those carrying the **X/*58:01 HLA-B* genotype, 4.51% had the *GG* genotype, and 7.52% had the *GA* genotype.

## 4. Discussion

This is the first study to determine the genotype and allele frequencies of the *HLA-B*58:01 HLA-B* gene using sequencing methods and to investigate its association with paraclinical characteristics and the SNP rs9263726 of the *PSORS1C1* gene in randomly selected gout patients living in Northeast Vietnam. Our study found a strong male predominance in gout cases (93.2%) and an older average age among female patients (70.33 years vs. 51.44 years for males), consistent with global epidemiological data. The proportion of female patients was very low, accounting for only 6% and 12.3% in gout patient groups from northern and central provinces of Vietnam, respectively [35,36]. Several review studies on gout have shown that gout is more common in men than in women, with a male-to-female ratio ranging from 3:1 to 10:1 [22,37]. Similarly, Zhu et al. (2011) noted that the incidence of gout is significantly higher in men and increases with age in both sexes, with a sharper rise in women after menopause, reflecting the protective effect of estrogen on serum uric acid levels [38]. A population-based study by Kuo et al. (2015) in the UK further corroborated this trend, showing that the incidence of gout among women rises significantly with age, especially after 60 years, narrowing the gender gap in older age groups [39]. Moreover, Evan and colleagues suggested that women with gout tend to be older, often presenting after age 60 years, aligning with the current study’s observation that most female patients were in the ≥60 age group [40]. Several studies showed that alcohol intake is strongly associated with an increased risk of gout and recurrent gout attacks [41,42]. Moreover, women tend to consume less alcohol and experience fewer alcohol-related issues compared to men. They also appear to be less susceptible to alcohol-related health risks [43]. Vietnam is considered a country with high alcohol consumption, particularly among men. A recent investigation reported that nearly 60% of surveyed individuals consumed alcohol, with approximately 50% of men drinking at a moderate level or higher [44]. In addition, our previous study showed that all randomly selected alcoholic cirrhosis patients living in Northeast Vietnam were male [33]. Therefore, we suggest that the higher prevalence of gout in men compared to women and the older age of female patients compared to males in this study may be related to alcohol consumption. Further research is needed to clarify the association between gender, alcohol consumption, and the risk of gout in Vietnam.

This study found weak but statistically significant positive correlations between uric acid and both PLT count (*r* = 0.174, *p* = 0.045) and serum Cre (*r* = 0.195, *p* = 0.025) in gout patients. In male gout patients, stronger and statistically significant positive correlations were observed between serum Ure levels and platelet count (PLT; *r* = 0.202, *p* = 0.024), urea (*r* = 0.217, *p* = 0.016), and creatinine (Cre; *r* = 0.215, *p* = 0.017). Tayefi et al. (2018) found an independent association between platelet count and uric acid levels in hypertensive individuals, suggesting a potential role of inflammatory or vascular processes in uric acid elevation [45]. Furthermore, Nishida (1992) reported a positive correlation between 24 h urinary creatinine and uric acid excretion in both gout patients and healthy subjects, supporting the connection between renal function and uric acid regulation [46]. Similarly, Ephraim et al. (2021) showed that serum uric acid is a more reliable marker of renal impairment than the uric acid-to-creatinine ratio in type 2 diabetes mellitus patients [47]. These global findings reinforce the notion that platelet counts and creatinine levels may serve as valuable indicators in understanding the pathophysiology of gout and its systemic implications. Elevated levels of HDL-C have been proposed to exert anti-inflammatory effects and modulate systemic inflammatory responses [48,49] while also being associated with a reduced risk of cardiovascular mortality [50,51]. Conversely, in female patients, the only significant association was a strong inverse correlation between Uri and HDL-C (*r* = −0.885, *p* = 0.002). The strong inverse relationship between uric acid and HDL-C in females warrants further investigation, as it may reflect gender-specific cardiovascular risk patterns associated with gout.

The allele frequency database “http://www.allelefrequencies.net/hla6006a.asp (accessed on 15 August 2023)” and several studies indicate that the *HLA-B*58:01* allele has been previously reported in unrelated healthy Vietnam populations from Hanoi and the Kinh ethnic group [13,30,31]. For the first time in Vietnam, we report the frequency of the *HLA-B*58:01* allele in a randomly selected gout patient group living in the northeastern region. The obtained results showed that the frequency of *HLA-B*58:01* in gout patients was 6.02%, with 12.03% of individuals being heterozygous (**X/*58:01*) and no individuals being homozygous (**58:01/*58:01*). The genotype and frequencies of the *HLA-B*58:01* allele observed in this Vietnamese cohort of gout patients are consistent with data reported in Vietnam [13,30,31]. Similar frequencies have been reported in studies conducted in Thailand and China, where *HLA-B*58:01* frequencies range from 6% to 8% in the general population but increase markedly among patients experiencing severe allopurinol-induced adverse drug reactions [7,8]. The *HLA-B*58:01* allele is codominant; therefore, an individual needs only one copy of the *HLA-B*58:01* allele to be at high risk of developing SCARs when using allopurinol [6,7]. The *HLA-B*58:01* allele was not detected in female gout patients in our study. We observed a statistically significant elevation in white blood cell counts among *HLA-B*58:01* heterozygous individuals (**X/*58:01*) compared to those without the allele (**X/*X*) in male gout patients, while direct studies examining the relationship between *HLA-B*58:01* and white blood cell levels are limited. This immunological predisposition might contribute to heightened inflammatory responses, reflected in elevated blood cell counts. Further research is warranted to elucidate the mechanisms by which *HLA-B*58:01* may influence inflammatory markers and to confirm these findings in larger, diverse cohorts.

In addition, previous reports have suggested that *PSORS1C1* rs9263726 could serve as a surrogate marker for detecting *HLA-B*58:01* in the Vietnamese population, offering a cost-effective and simplified alternative to genetic screening. In this study, the observed genotype distribution of rs9263726 was 84.21% *GG* (wild type), 15.79% *GA* (heterozygous), and 0% *AA* (homozygous mutant), with allele frequencies of 92.11% *G* and 7.89% *A*, suggesting a low prevalence of the *A* allele in the study population. This pattern is consistent with findings in certain populations where the *A* allele is relatively uncommon. For instance, in an Australian cohort, the *GG* genotype was observed in 68.8% of individuals, *GA* was observed in 29.2%, and *AA* was observed in 2.0%, showing a higher frequency of the *A* allele than in this study [16]. Further analysis showed that the majority of individuals (79.7%) had the **X/*X* genotype for *HLA-B* along with the *GG* genotype for *PSORS1C1* rs9263726, while a smaller fraction (8.27%) possessed the **X/*X* and *GA* genotype combination. Among those carrying the **X/*58:01* genotype for *HLA-B*, 4.51% had the *GG* genotype, and 7.52% had the *GA* genotype. This distribution indicates a relatively weak link between *PSORS1C1* rs9263726 and *HLA-B*58:01* in this population. Similar patterns have been observed in other studies, where it was found that [16,17], although rs9263726 showed linkage disequilibrium with *HLA-B*58:01* in Han Chinese populations, the strength of this association varied, with not all *HLA-B*58:01* carriers showing the rs9263726-A allele. Conversely, in Han Chinese populations, studies have also reported a stronger linkage disequilibrium between rs9263726 and *HLA-B*58:01*, with the *A* allele serving as a more reliable surrogate marker for *HLA-B*58:01* [52]. However, this association is not consistent across all ethnic groups. Research in Tibetan and Hui populations demonstrated a weaker linkage disequilibrium between rs9263726 and *HLA-B*58:01*, limiting the utility of rs9263726 as a surrogate marker in these groups [52]. Therefore, the low frequency of the *A* allele in our study population suggests that rs9263726 may not be a reliable surrogate marker for *HLA-B*58:01*, and direct genotyping of *HLA-B**58:01 remains the most accurate method for identifying individuals at risk of allopurinol-induced SCARs.

Due to the limited sample size of our study, particularly the low number of female gout patients, further research in larger populations is needed to confirm these findings. Additionally, we have not yet investigated the association of *HLA-B*58:01* and *PSORS1C1* rs9263726 in patients undergoing uric acid-lowering treatment with allopurinol.

## 5. Conclusions

This is the first study to identify the allele and genotype frequencies of *HLA-B*58:01* using the Sanger sequencing method and to investigate its association with paraclinical characteristics and the SNP rs9263726 of the *PSORS1C1* gene in randomly selected gout patients living in Northeast Vietnam. Our investigation revealed a statistically significant difference in age between male and female patients. A significant positive correlation between platelet count, serum creatinine level, and uric acid concentration was revealed in male gout patients. The significant association was a strong inverse correlation between Uri and HDL-C. The *HLA-B*58:01* allele was not detected in female gout patients. White blood cell levels showed a statistically significant difference between **X*/**58:01* and **X/*X* groups in male gout patients. *HLA-B*58:01* was not consistently associated with the SNP rs9263726 of the *PSORS1C1* gene, suggesting that rs9263726 may not be a reliable surrogate marker for *HLA-B*58:01* in gout patients. Our results provide valuable scientific information for the development of genetic screening strategies for individuals carrying *HLA-B*58:01* in Vietnam.

## Figures and Tables

**Figure 1 diagnostics-15-02114-f001:**
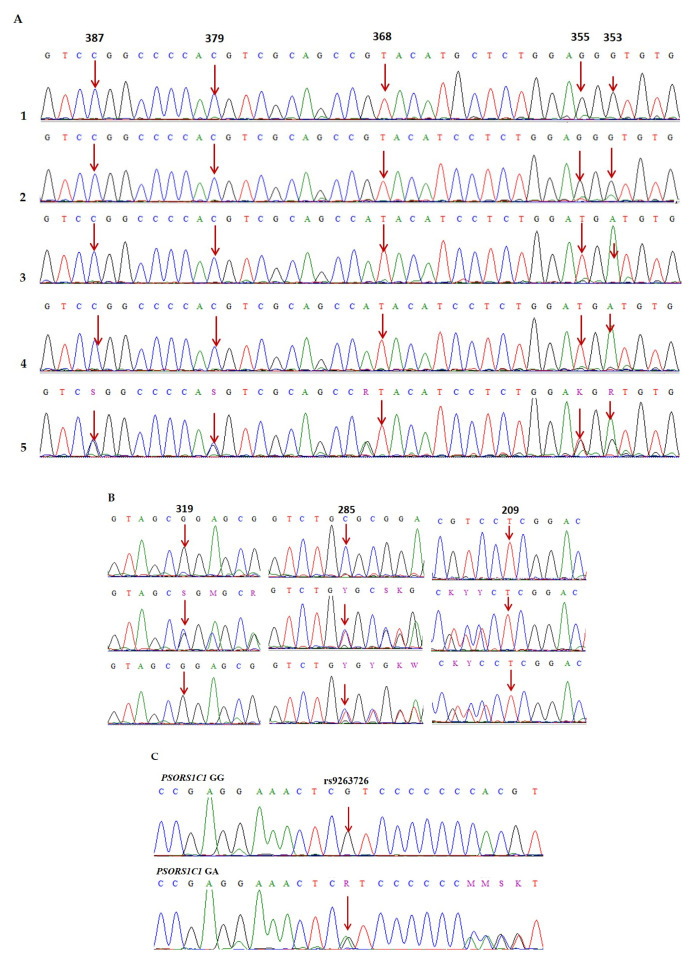
Partial Sanger sequencing chromatograms of exons 3 and 2 of the *HLA-B* gene (**A**,**B**) and exon 3 of the *PSORS1C1* gene (**C**). Abbreviations: Red arrows indicate the nucleotide positions used to identify *HLA-B*58:01* in exons 3 and 2 of the *HLA-B* gene as well as the rs9263726 SNP in the *PSORS1C1* gene (*PSORS1C1 GG*: homozygous wild-type genotype; *PSORS1C1 GA*: heterozygous mutant genotype). Heterozygous nucleotide cases (S—GC, R—GA, K—GT, M—CA, Y—CT, K—GT, W—AT).

**Table 1 diagnostics-15-02114-t001:** Primers used for *HLA-B* (exons 2 and 3) and *PSORS1C1* (exon 3) fragment amplification and sequencing.

Gene Region	Forward Primer (5′–3′) ^a^	Reverse Primer (5′–3′) ^b^	Fragment Size (bp)
*HLA-B* exon 2	CAGTTCTAAAGTCCCCACGCAC	GATCTCGGACCCGGAGACTC	613
*HLA-B* exon 3	AGGCGC GTTTACCCGGTTTC	CATTCAACGGAGGGCGACATTC	495
*PSORS1C1* exon 3	CTAGCTTTGTCCTCAGGCCAAC	AGAAGGTGCATCTGGCTCACC	265

^a,b^ Primers used for sequencing.

**Table 2 diagnostics-15-02114-t002:** Age and gender characteristics of study subjects.

Age (year)	Gender		Total
	Male	Female	
≤40	35 (100.0%)	0 (0.0%)	35 (26.3%)
41 ≤ 59	54 (96.4%)	2 (3.6%)	56 (42.1%)
≥60	35 (83.3%)	7 (16.7%)	42 (31.6%)
Total	124 (93.2%)	9 (6.8%)	133 (100.0%)
Average age	51.44 ± 14.59	70.33 ± 10.64	52.71 ± 15.09
*p* value	<0.001		

Abbreviations: *p* < 0.05 was considered statistically significant.

**Table 3 diagnostics-15-02114-t003:** Correlation between uric acid concentration and paraclinical characteristics.

Paraclinical Characteristics	Gout Patients (N = 133)		Male Gout Patients (N = 124)		Female Gout Patients (N = 9)	
	r	*p* Value	r	*p* Value	r	*p* Value
RBC (10^12^/L)	−0.0006	0.946	−0.033	0.719	0.389	0.301
HGB (g/L)	−0.064	0.465	−0.101	0.265	0.325	0.393
HCT (%)	−0.061	0.487	−0.107	0.238	0.406	0.278
WBC (10^9^/L)	0.013	0.881	−0.006	0.945	0.331	0.384
NE (%)	−0.027	0.757	−0.058	0.526	0.223	0.564
LYM (%)	0.074	0.396	0.063	0.487	0.359	0.342
PLT (10^12^/L)	0.174	0.045	0.202	0.024	−0.110	0.778
Glu (mmol/L)	−0.038	0.663	−0.063	0.487	0.572	0.107
Ure (µmol/L)	0.159	0.067	0.217	0.016	−0.330	0.386
Cre (µmol/L)	0.195	0.025	0.215	0.017	−0.001	0.997
TC (mmol/L)	−0.045	0.603	−0.087	0.337	0.605	0.084
TG (mmol/L)	−0.079	0.366	−0.101	0.263	0.298	0.436
HDL-C (mmol/L)	0.025	0.772	−0.046	0.612	0.885	0.002
LDL-C (mmol/L)	−0.034	0.701	−0.040	0.656	0.213	0.582

Abbreviations: N, number of subjects; RBC, red blood cell; HBG, hemoglobin; HCT, hematocrit; WBC, white blood cell; NE, neutrophil; LYM, lymphocyte; PLT, platelet; Glu, glucose; Ure, Urea; Cre, creatinine; TG, triglyceride; TC, total cholesterol; HDL-C, high-density lipoprotein cholesterol; LDL-C, light-density lipoprotein cholesterol; r, correlation coefficient. *p* < 0.05 was considered statistically significant.

**Table 4 diagnostics-15-02114-t004:** Allele and genotype frequencies of *HLA-B*58:01* and *PSORS1C1* SNP rs9263726.

Gene	Polymorphism	Nucleotide Change	Genotypes and Alleles	N and n	Frequencies (%)
*HLA-B* exons 2 and 3		c.209A, 285A>G, 319G>C, 353C>T, 355C>A, 368A, 379G>C, 387G>C	**X/*X*	117	87.97
			**X/*58:01*	16	12.03
			**58:01/*58:01*	0	0
			**X*	249	93.98
			**58:01*	16	6.02
*PSORS1C1 exon 3*	rs9263726	c.1418G>A	GG	112	84.21
			GA	21	15.79
			AA	0	0
			G	245	92.11
			A	21	7.89

Abbreviations: N, number of subjects; n is the number of alleles; **X* represents any *HLA-B* allele other than *HLA-B*58:01* (**58:01*); *GG*: *PSORS1C1* homozygous wild-type genotype; *GA*: *PSORS1C1* heterozygous mutant genotype.

**Table 5 diagnostics-15-02114-t005:** Correlation between **X/*X* and *X*/*58:01* and paraclinical characteristics.

Paraclinical Characteristics	Genotypes	Gout Patients (N = 133)		Male Gout Patients (N = 124)	
		Mean ± SD	*p* Value	Mean ± SD	*p* Value
RBC (10^12^/L)	**X/*X*	5.120 ± 1.356	0.979	5.1982 ± 1.37058	0.845
	**X/*58:01*	5.128 ± 0.949		5.1288 ± 0.94898	
HGB (g/L)	**X/*X*	138.11 ± 18.348	0.305	139.51 ± 18.160	0.458
	**X/*58:01*	143.06 ± 15.303		143.06 ± 15.303	
HCT (%)	**X/*X*	41.188 ± 4.857	0.102	41.597 ± 4.730	0.178
	**X/*58:01*	43.344 ± 5.316		43.344 ± 5.316	
WBC (10^9^/L)	**X/*X*	9.649 ± 3.036	0.018	9.617 ± 2.963	0.018
	**X/*58:01*	12.351 ± 9.275		12.351 ± 9.275	
NE (%)	**X/*X*	48.277 ± 27.121	0.632	50.023 ± 26.214	0.811
	**X/*58:01*	51.669 ± 21.459		51.669 ± 21.459	
LYM (%)	**X/*X*	19.097 ± 13.672	0.877	20.054 ± 13.565	0.908
	**X/*58:01*	19.646 ± 10.084		19.646 ± 10.084	
PLT (10^12^/L)	**X/*X*	274.315 ± 74.725	0.757	274.646 ± 72.420	0.764
	**X/*58:01*	280.366 ± 59.102		280.366 ± 59.102	
Glu (mmol/L)	**X/*X*	6.313 ± 2.855	0.288	6.352 ± 2.943	0.327
	**X/*58:01*	7.098 ± 1.805		7.098 ± 1.805	
Ure (µmol/L)	**X/*X*	6.542 ± 3.337	0.228	6.353 ± 3.249	0.313
	**X/*58:01*	5.513 ± 1.682		5.513 ± 1.682	
Cre (µmol/L)	**X/*X*	104.171 ± 36.676	0.535	103.316 ± 36.799	0.598
	**X/*58:01*	98.339 ± 19.912		98.339 ± 9.912	
Uri (µmol/L)	**X/*X*	522.868 ± 92.849	0.788	524.669 ± 93.033	0.733
	**X/*58:01*	516.348 ± 74.586		516.348 ± 74.586	
TC (mmol/L)	**X/*X*	5.102 ± 0.999	0.813	5.080 ± 0.991	0.874
	**X/*58:01*	5.034 ± 1.525		5.034 ± 1.525	
TG (mmol/L)	**X/*X*	2.792 ± 1.680	0.927	2.837 ± 1.719	0.991
	**X/*58:01*	2.832 ± 1.258		2.832 ± 1.258	
HDL-C (mmol/L)	**X/*X*	1.295 ± 0.328	0.216	1.279 ± 0.313	0.275
	**X/*58:01*	1.189 ± 0.274		1.189 ± 0.274	
LDL-C (mmol/L)	**X/*X*	2.818 ± 0.837	0.157	2.807 ± 0.859	0.183
	**X/*58:01*	2.494 ± 0.978		2.494 ± 0.978	

Abbreviations: N, number of subjects; RBC, red blood cell; HBG, hemoglobin; HCT, hematocrit; WBC, white blood cell; NE, neutrophil; LYM, lymphocyte; PLT, platelet; Glu, glucose; Ure, Urea; Cre, creatinine; Uri, uric acid; TG, triglyceride; TC, total cholesterol; HDL-C, high-density lipoprotein cholesterol; LDL-C, light-density lipoprotein cholesterol; **X*, any *HLA-B* allele other than *HLA-B*58:01* (**58:01*); SD, standard deviation. *p* < 0.05 was considered statistically significant.

**Table 6 diagnostics-15-02114-t006:** Genotype frequencies of *HLA-B* and the *PSORS1C1* SNP rs9263726 in combination.

Genotype		N	(%)
*HLA-B*	*PSORS1C1* (rs9263726)	133	100
**X/*X*	GG	106	79.7
**X/*X*	GA	11	8.27
**X/*58:01*	GG	6	4.51
**X/*58:01*	GA	10	7.52

Abbreviations: N, number of subjects; **X*, any *HLA-B* allele other than *HLA-B*58:01*; *GG*: *PSORS1C1* homozygous wild-type genotype; *GA*: *PSORS1C1* heterozygous mutant genotype.

## Data Availability

The original contributions presented in this study are included in the article. Further inquiries can be directed to the corresponding author.

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
