# Peer review of "Genotype Frequency of HLA-B*58:01 and Its Association with Paraclinical Characteristics and PSORS1C1 rs9263726 in Gout Patients"

_diagnostics, 2025, doi:10.3390/diagnostics15162114_

Round 1
Reviewer 1 Report
Comments and Suggestions for Authors
I appreciated having the opportunity to review this article about the frequency of the HLA-B58:01 allele in gout patients from Northeast Vietnam and its potential correlation with the SNP rs9263726 in the PSORS1C1 gene, as well as with selected paraclinical characteristics.
I just have some minor suggestions that I think can help improve the paper overall.
The authors should abbreviate the abstract as it is overly condensed and they moreover should more clearly define the research aims, methods, and key results.
it is very important to be able to reproduce the study. For this reason, please include more detailed parameters for the PCR and sequencing procedures.
The discussion currently restates the results but in may opinion it would be good to deepen their interpretation. The authors should elaborate on: why rs9263726 is a poor marker in this population and they should state more specifically limitations of the study (e.g., sample size, single-center recruitment, absence of control group).
Author Response
- The authors should abbreviate the abstract as it is overly condensed and they moreover should more clearly define the research aims, methods, and key results.
Answer: We agree with this comment. Therefore, we have revised the research aims, methods, and key results. These changes highlighted and can be found in page 1, pragraph 1 and lines 14-37 of the manuscript.
- It is very important to be able to reproduce the study. For this reason, please include more detailed parameters for the PCR and sequencing procedures.
Answer: We agree with this comment. We have added and highlighted the information in page 4, pragraph 3 and lines 133-136; pragraph 3 and lines 141- 150 of the manuscript.
- The discussion currently restates the results but in may opinion it would be good to deepen their interpretation. The authors should elaborate on: why rs9263726 is a poor marker in this population and they should state more specifically limitations of the study (e.g., sample size, single-center recruitment, absence of control group).
Answer: We agree with this comment. We conducted additional analyses to clarify the association between uric acid levels and clinical characteristics in both male and female patient groups. These changes highlighted and can be found in table 3, page 6, pragraph 1 and lines 190-195; page 11, paragraph 2 and lines 294-297, 307-313. Furthermore, the relationship between the HLA-B*58:01 carriers and paraclinical characteristics in male patients was also examined. This change can be found in table 5, page 11, paragraph 3 and lines 327-330. The study's limitations have also been added and emphasized on page 12, paragraph 3, and lines 363-366.
The rs9263726 variant is easier to identify than HLA-B*58:01, an allele variant strongly associated with SCAR when treating gout with allopurinol. Our study aimed to investigate the combination of HLA-B*58:01 and rs9263726 to determine whether rs could serve as a surrogate marker for HLA-B*58:01. Therefore, we did not focus on an in-depth analysis of rs9263726.
Reviewer 2 Report
Comments and Suggestions for Authors
This study examined the frequency of the HLA-B58:01 allele and its link to clinical markers and the rs9263726 SNP in 133 gout patients from Northeast Vietnam. The allele was found in 6% of patients, and some paraclinical differences, such as white blood cell count, were noted between carriers and non-carriers. However, rs9263726 did not reliably predict HLA-B58:01 presence, suggesting it is not a suitable surrogate marker in this population. The authors note that further research is needed to clarify genotype-related clinical patterns.
Major Comments:
- The focus is on the rs9263726 SNP, a missense change in PSORS1C1. This same SNP is strongly associated with a long list of diseases, and such pleiotropic effects were not mentioned (celiac disease, hypothyroidism, psoriasis, dermatitis and eczema, and more). Therefore, avoid confusion to only connet it to gout.
- Is there any data from Northeast Vietnam for non-gout patients (controls)? Including such data could provide a valuable reference point for this study. The negative result regarding the SNP's relevance is certainly important, but it may also be sensitive to the limited cohort size. Validation of allele frequencies in larger populations might be informative (e.g., FinnGen, UK Biobank, All of Us).
- The cohort is male-dominated (as stated). It would be useful to repeat the analysis with a separate column (e.g., in Table 3) showing results for males only, given that only 9 of 133 participants are female—and they differ in age, potentially masking male-specific signals.
- While the motivation for this study is well established and the background is clearly explained, it is also important to direct readers to the accumulated knowledge on rs9263726. Resources such as Open Targets (with evidence on drug relevance), UniProtKB (for polymorphic protein sites), and dbSNP (for global allele frequency data) and PharmGKB are essential. Including this information in the introduction, discussion, or as a summary table would help contextualize the findings against global data, especially given the small cohort studied.
- There is some repetition in the writing. For example, results in Tables 2 and 3 are thoroughly described in free text (lines 126–146), which could be condensed for clarity. The conclusion section also repeats many of these points. A more concise summary would improve readability.
- The discussion is somewhat disjointed. While the introduction centers on allopurinol response, the discussion shifts focus to weak (though significant) associations between platelet count, creatinine, and uric acid—hinting at vascular or inflammatory processes. It then moves away again to alcohol consumption in men as a key gout risk factor. These transitions need better justification or clarification of relevance.
Minor Comments:
- Please change “Viet Nam” to “Vietnam” in the affiliation.
- In Figure 1, explain the ‘nucleotide’ annotations M and R, which represent combinations of nucleotides.
- Missing key references. For example, a careful study in Thailand confirmed that the rs9263726-HLA-B*58:01 linkage is not fully consistent and therefore not clinically useful (e.g., https://doi.org/10.1155/2017/2738784). This should be cited.
Author Response
Major Comments
- The focus is on the rs9263726 SNP, a missense change in PSORS1C1. This same SNP is strongly associated with a long list of diseases, and such pleiotropic effects were not mentioned (celiac disease, hypothyroidism, psoriasis, dermatitis and eczema, and more). Therefore, avoid confusion to only connet it to gout.
- While the motivation for this study is well established and the background is clearly explained, it is also important to direct readers to the accumulated knowledge on rs9263726. Resources such as Open Targets (with evidence on drug relevance), UniProtKB (for polymorphic protein sites), and dbSNP (for global allele frequency data) and PharmGKB are essential. Including this information in the introduction, discussion, or as a summary table would help contextualize the findings against global data, especially given the small cohort studied.
Answer: We believed that the rs9263726 variant is easier to identify than HLA-B*58:01, an allele variant strongly associated with SCAR when treating gout with allopurinol. Our study aimed to investigate the combination of HLA-B*58:01 and rs9263726 to determine whether rs9263726 could serve as a surrogate marker for HLA-B*58:01. Therefore, we did not focus on an in-depth analysis of rs9263726.
- Is there any data from Northeast Vietnam for non-gout patients (controls)? Including such data could provide a valuable reference point for this study. The negative result regarding the SNP's relevance is certainly important, but it may also be sensitive to the limited cohort size. Validation of allele frequencies in larger populations might be informative (e.g., FinnGen, UK Biobank, All of Us).
Answer: To date, there are no available data on HLA-B*58:01 in both non-gout and gout patients in the Northeast region of Vietnam. We have added additional information to clarify this issue. The additions can be found on page 11, We analyzed and compared the frequency of the HLA-B*58:01 allele obtained in our study with existing allele frequency databases. The revisions and additions can be found on page paragaraph 3 and lines 314-317.
- The cohort is male-dominated (as stated). It would be useful to repeat the analysis with a separate column (e.g., in Table 3) showing results for males only, given that only 9 of 133 participants are female—and they differ in age, potentially masking male-specific signals.
Answer: We agree with this comment. We have added columns analyzing the association between uric acid levels and clinical characteristics in both female and male patients. The added columns are highlighted and can be found on page 5 table 3.
- There is some repetition in the writing. For example, results in Tables 2 and 3 are thoroughly described in free text (lines 126–146), which could be condensed for clarity. The conclusion section also repeats many of these points. A more concise summary would improve readability.
Answer: We agree with this comment. We have shortened the descriptions of Tables 2 and 3 to avoid redundancy with the tables themselves.
- The discussion is somewhat disjointed. While the introduction centers on allopurinol response, the discussion shifts focus to weak (though significant) associations between platelet count, creatinine, and uric acid—hinting at vascular or inflammatory processes. It then moves away again to alcohol consumption in men as a key gout risk factor. These transitions need better justification or clarification of relevance.
Answer: We agree with this comment. We have revised the discussion section to better clarify the research findings; the modifications highlighted and can be found on page 10, paragraph 2 and lines 262-266, 274-284; page 11, paragraph 2 and lines 287-290, 300-310.
Minnor Comments
- Please change “Viet Nam” to “Vietnam” in the affiliation.
Answer: Agree. We have revised and highlight in line 13
- In Figure 1, explain the ‘nucleotide’ annotations M and R, which represent combinations of nucleotides.
Answer: Agree. We have added information to explain the ‘nucleotide’ annotations M and R…, which represent combinations of nucleotides. This change can be found in page 7, lines 212-213
- Missing key references. For example, a careful study in Thailand confirmed that the rs9263726-HLA-B*58:01 linkage is not fully consistent and therefore not clinically useful (e.g., https://doi.org/10.1155/2017/2738784). This should be cited.
Answer: We agree with this comment. We have added references related to the introduction and discussion. The added references have highlighted and can be found in page 14-16 with lines 436-438, 448-467, 475-478, 490-491 and 504-513.
Reviewer 3 Report
Comments and Suggestions for Authors
Comments:
- In the abstract section, please start the line with "The HLA-23 B*58:01 allele frequency" findings.
- Please elaborate on the methodology section.
- In the chromatogram images, please add the reference images and highlight the changes by comparing with the reference sequences.
- Please compare the present findings with the other studies.
- State the limitations of the study.
Author Response
- In the abstract section, please start the line with "The HLA-23 B*58:01 allele frequency" findings.
Answer: We suggest that we should use "The HLA-B*58:01 allele" instead of "The HLA-B*58:01 allele frequency" because the HLA-B*58:01 allele is strongly linked to severe cutaneous adverse reactions (SCARs) during allopurinol treatment.
- Please elaborate on the methodology section.
Answer: We agree with this comment. We have added information about the methodology in pagae 3, lines 132-136 and page 4 and lines 140-150.
- In the chromatogram images, please add the reference images and highlight the changes by comparing with the reference sequences.
Answer: We agree with this comment. We have added reference images for sequencing chromatograms of exon 3 HLA-B gene in Figure 1 (1A). Additionally, we have provided complete captions to ensure clarity and accuracy. This change can be found in figure 1 page 7 and lines 207-213.
- Please compare the present findings with the other studies.
Answer: We agree with this comment. We have revised the discussion section to better clarify the research findings; the modifications highlighted and can be found on page 10, paragraph 2 and lines 262-266, 274-284; page 11, paragraph 2 and lines 287-290, 300-310.
- State the limitations of the study.
Answer: We agree with this comment. The study's limitations have also been added and emphasized on page 12, paragraph 3, and lines 363-366.
Reviewer 4 Report
Comments and Suggestions for Authors
Dear Authors,
First of all, congratulations for your interesting work. I hope that my hints will help you in the next steps of improvement and the final manuscript will be really valuable for the readers. Although the research is not very innovative and performed on a very confined society (why exactly these people? have you describe this in the paragraph about limitations of the study? considered inbred too? also, there is significantly more men than women, why? ---> lines 132 - 134 can be different if you had collected similar number of male/female patients, although lines 214-216 explaines a bit this phenomenon), about only one selected variant, it might be interesting and useful especially among pharmacogenomic field.
There are some minor English mistakes, please correct the grammar.
Can you please think of any graphs and graphical representation of the results in your research? It would definitely facilitate the reading process.
Finally, I would like to encourage you to come back to the abstract part of your manuscript. In a current form it looks like a conference note or an abstract of a poster, it is not encouraging readers to dig deeper into your research, it seems not interesting. An abstract should have more popular-science style, to show the importance and significance of your work, not solely a mini-summary of what you have done. I strongly encourage you to rewrite this section.
Author Response
- Why exactly these people? have you describe this in the paragraph about limitations of the study? considered inbred too? also, there is significantly more men than women, why? ---> lines 132 - 134 can be different if you had collected similar number of male/female patients, although lines 214-216 explaines a bit this phenomenon), about only one selected variant, it might be interesting and useful especially among pharmacogenomic field.
Answer: We agree with this comment. The gout patient group in this study had no familial relationships and was randomly selected. The proportion of female patients in the gout group was very low, which is consistent with the epidemiological characteristics of gout in Vietnam. Some additional revisions can be found on page 3, paragraph 4, line 113. The study's limitations have also been added and emphasized on page 12, paragraph 3, and lines 363-366.
- There are some minor English mistakes, please correct the grammar.
Answer: We agree with this comment. We have revised all the mistakes and grammar carefully
- Can you please think of any graphs and graphical representation of the results in your research? It would definitely facilitate the reading process.
Answer: We agree with this comment. We have added information about the methodology in page 4, paragraph 4 and lines 140-150. Additionally, we have revised the figure and provided complete captions to ensure clarity and accuracy. This change can be found in figure 1 page 7 and lines 207-213.
- Finally, I would like to encourage you to come back to the abstract part of your manuscript. In a current form it looks like a conference note or an abstract of a poster, it is not encouraging readers to dig deeper into your research, it seems not interesting. An abstract should have more popular-science style, to show the importance and significance of your work, not solely a mini-summary of what you have done. I strongly encourage you to rewrite this section.
Answer: We agree with this comment. We have rewritten the abstract to highlight the importance and significance of our stuudy. These changes highlighted and can be found in page 1, pragraph 1 and lines 14-37 of the manuscript.
Round 2
Reviewer 2 Report
Comments and Suggestions for Authors
The authors addressed most points in a satisfactory way. The first reference is in Vietnamese, it is less appropriate for an international journal.
Other Referneces that are not a full paper (but a link to data) are better added as a link to Methods or to a data availability section (like 1-6, 14)
Author Response
Thank you for your suggestions that greatly help to improve the quality of our manuscript. We agree with all your comments, and we answered and corrected point by point the manuscript as shown below.
- The first reference is in Vietnamese, it is less appropriate for an international journal.
Answer: We agree with this comment. We have replaced this reference with an English-language source và highlighted lines 398-399 of the manuscript.
- Other Referneces that are not a full paper (but a link to data) are better added as a link to Methods or to a data availability section (like 1-6, 14)
Answer: We agree with this comment. References 2–5 can be cited either as website sources or as parts of books. Therefore, we have revised the citation format and highlighted it in the lines 400-411 of the manuscript.
After carefully considering the relevance of the reference to the presented content, we have omitted Reference No. 6 because it is a book containing parts of References 2–5.
As for References No. 14 and 31, we have cited the links to the data sources in the manuscript, and they have been highlighted in lines 74, 107, 145 and 310 of the manuscript.
Reviewer 3 Report
Comments and Suggestions for Authors
The authors have revised the manuscript significantly.
Author Response
The authors have revised the manuscript significantly.
Answer: We thank your kind reconsideration of our revised manuscript.